

# Inhibition of hepatocellular carcinoma growth *via* modulation of the miR-221/SOX11 axis by curcumin and berberine

Sheng Li[1], Xiaoliang Cai[1], Liang Chen[2], Manbian Lin[2], Ziqi Zhu[2], Huihuang Xiao[2], Pingping Nie[2], Quanwen Chen[2] and Xiaoyu Yang[3]

[1] Department of Internal Medicine, Fuzhou Hospital of Traditional Chinese Medicine, Fuzhou, China
[2] Department of Medical Oncology, Fuzhou Hospital of Traditional Chinese Medicine, Fuzhou, China
[3] Fuzhou Hospital of Traditional Chinese Medicine, Fuzhou, China

Corresponding authors
Sheng Li, fzszyyls@tom.com
Xiaoyu Yang, yxyfj@163.com

## ABSTRACT

Hepatocellular carcinoma (HCC) is a fatal malignancy that has limited treatment options. This study focused on the potential therapeutic effects of curcumin (CUR) and berberine (BBR) on the miR-221/SRY-box transcription factor 11 (SOX11) axis in HCC. We investigated the combined effects of CUR and BBR on HEPG2 and Huh7 cell survival and miR-221 expression using Cell Counting Kit-8 assays and RT-qPCR, respectively. Western blotting was used to detect changes in the apoptosis-related caspase-3/9 protein levels. We performed bioinformatics analysis and dual-luciferase assays and measured apoptotic protein levels to assess the role of the miR-221/SOX11 axis in mediating the effects of CUR-BBR. Both CUR and BBR suppressed HCC cell growth in a dose-dependent manner, with the most potent combined effect observed at a 2:1 ratio. CUR-BBR treatment significantly downregulated miR-221 expression, and miR-221 overexpression partially reversed the CUR-BBR-mediated decrease in cell survival. In addition, SOX11 was found to be a direct target of miR-221. CUR-BBR treatment upregulated SOX11 expression, and overexpression of SOX11 restored the inhibitory effects of CUR-BBR on cell growth, migration, and invasion and promoted apoptosis in the presence of miR-221. Furthermore, CUR-BBR activated pro-apoptotic proteins caspase-3/9 through the miR-221/SOX11 axis. The combined effect of CUR-BBR played an important role in inhibiting the growth of HCC cells. This combined effect was achieved by regulating the miR-221/SOX11 axis and activating the synthesis of pro-apoptotic proteins. Our findings highlight a promising combined therapeutic approach for HCC and underscore the importance of targeting the miR-221/SOX11 axis.

# INTRODUCTION

Hepatocellular carcinoma (HCC) is a major global health issue that accounts for a significant proportion of cancer-related mortalities worldwide (*Nagaraju et al., 2022*). Despite poor prognosis, late-stage diagnosis, and limited therapeutic options, the management of HCC remains challenging (*Wen et al., 2022*). Molecular therapies targeting

specific biological pathways implicated in HCC development are currently being explored as potential novel treatment modalities. Better understanding of the molecular mechanisms underlying HCC is critical for the development of targeted therapies.

Although HCC is a major global health concern, emerging research has identified potential therapeutic agents, among which, curcumin (CUR) and berberine (BBR) have shown promise. CUR, a polyphenol extracted from turmeric, has antioxidant, anti-inflammatory, antibacterial, and antiviral properties (*Chamani et al., 2022*). Alone or in combination with other drugs, it may be effective in cancer treatment (*Ziasarabi, Sahebkar & Ghasemi, 2021*). BBR, a quaternary ammonium alkaloid isolated from *Coptis coptidis*, acts as a tumor suppressor in various cancer cells (*Rauf et al., 2021*). Both CUR and BBR induce significant apoptosis in HCC cells, showing positive effects on HCC treatment (*Wu et al., 2022a*). In recent years, many studies have shown that CUR and BBR can produce synergistic effects that are more effective in killing cancer cells, such as breast cancer and glioblastoma (*Maiti, Plemmons & Dunbar, 2019*; *Ziasarabi, Sahebkar & Ghasemi, 2021*). However, whether there is a synergistic effect and the underlying molecular mechanism of the combined treatment in HCC have not been fully elucidated. In this study, two cell lines, HEPG2 and Huh7, which are commonly used to detect the effects of drug therapy on HCC (*Huang et al., 2021b*; *Wu et al., 2022b*), were used to explore the synergistic effects of CUR and BBR.

One such molecular mechanism involves microRNAs (miRNAs), the small non-coding RNAs that play crucial roles in the post-transcriptional regulation of gene expression in HCC (*Hu et al., 2021*). Research has shown that miR-498 promotes HCC metastasis (*Lv et al., 2021*) and miR-18a-5p promotes HCC cell survival (*Cui et al., 2021*). Among these, miR-221 plays a critical role in several human malignancies. In HCC tissues, miR-221 high expression has been consistently associated with poor prognosis and aggressive tumor behavior (*Di Martino et al., 2022*). miRNAs can also inhibit the transcriptional activity and translation of the target by binding to the 3′ untranslated region (UTR) of the downstream target, thereby interfering with the occurrence and development of HCC (*Zhang et al., 2021a*). miR-221 can affect the progression of HCC through downstream targets such as DAB2 interacting protein or LIF receptor subunit alpha (*Liu et al., 2019b*; *Tan et al., 2022*). In this study, we aimed to investigate the combined effect of CUR and BBR on HCC cells and discern the potential role of miR-221 in mediating these effects.

Several malignancies are frequently abnormally expressed in the SRY-box transcription factor (SOX) family, which belongs to group C of the SRY-related high-mobility box family (*Fu et al., 2022*). The major members of the SOX family are SOX1, SOX4, and SOX11 (*Wan et al., 2019*). Among these, SOX11 has a cancer-suppressive effect on a variety of cancers, inhibits the migration of colon cancer cells (*Huang et al., 2022*), and excessive SOX11 alleviates HCC progression (*Liu et al., 2019c*). Previous studies have shown that SOX11 is maintained at low levels in HCC, and its overexpression significantly inhibits HCC progression by promoting apoptosis and cell cycle arrest (*Liu et al., 2019c*).

Cysteinyl aspartate-specific proteinase caspase (caspase) is one of the main enzymes that trigger apoptosis. When caspases are activated, a series of apoptotic proteases are released, ultimately causing cells to undergo irreversible apoptosis. The activation of

**Table 1 Sequences of miR-221 mimic, inhibitor, and their negative control.**

| miRNA | Sequences 5′–3′ |
| --- | --- |
| Mimic | AGCTACATTGTCTGCTGGGTTTC |
| Mimic NC | TCTGGTGCCGATTTAGTTGTACC |
| Inhibitor | GAAACCCAGCAGACAATGTAGCT |
| Inhibitor NC | CAGTACTTTTGTGTAGTACAA |

caspase-3/9, major members of the caspase family, is essential for the death of tumor cells (*Qiu et al., 2020*), thereby effectively preventing HCC progression (*Ni, Yuan & Wu, 2021*).

In this study, we attempted to understand the mechanism of action of CUR-BBR combination therapy through the interaction between miR-221 and SOX11 during the growth and apoptosis of HCC cell lines. Our findings highlight a promising combined therapeutic approach for HCC and underscore the importance of targeting the miR-221/SOX11 axis.

# METHODS

## Cell culture and transfection

Procell (Wuhan, China) provided HEPG2 and Huh7 cells, which were maintained at 37 °C and 5% $CO_2$. Huh7 and HEPG2 cells were maintained in Dulbecco's modified Eagle's medium (Gibco, Thermo Fisher Scientific, Inc., Waltham, MA, USA) and minimum essential medium, respectively, both medium containing 10% fetal bovine serum (FBS, Gibco). Both HEPG2 and Huh7 cells are derived from human HCCs and are widely acknowledged as representative models for *in vitro* HCC studies. By utilizing both cell lines, this study aimed to draw a comprehensive understanding, accounting for potential variability and ensuring broader applicability of the findings. HEPG2 and Huh7 cells used in this study were between passages 5–10. The cells were seeded at a density of 5 x $10^4$ cells per well for all experiments. CUR (Cat No. C1386; ≥95% HPLC) and BBR (Cat No. 08511; ≥98% HPLC) were purchased from Sigma-Aldrich (MO). Huh7 and HEPG2 cells were exposed to varying concentrations of CUR and BBR (0, 3.125, 6.25, 12.5, 25, 50, 100, and 200 μM), both individually and in combination at ratios of 1:1, 1:2, 1:4, 2:1, and 4:1 for 48 h. Lipofectamine® 2000 (Invitrogen, Waltham, MA, USA) miR-221 mimic/inhibitor, overexpression (ov) SOX11 plasm, and their negative controls were purchased from GenePharma Biotechnology Co., Ltd. (Shanghai, China), and were transfected into HEPG2 and Huh7 cells using Lipofectamine® 2000 (Invitrogen, Waltham, MA, USA) for 4 h, and their miR-221 mimic/inhibitor sequences are listed in Table 1.

## Cell survival assay

Both HCC cell lines were exposed to CUR and BBR for 24 and 48 h, respectively. Cell Counting Kit-8 (CCK8; Beyotime, Shanghai, China) solution was added and incubated at 37 °C for 2 h. The absorbance was recorded at 450 nm using a microplate reader (Thermo Fisher Scientific), and the half-maximal inhibitory concentration (IC50) was evaluated to determine the optimal CUR-BBR ratio.

**Table 2 Primer sequences for RT-qPCR.**

| Gene | Forward 5′–3′ | Reverse 5′–3′ |
| --- | --- | --- |
| β-actin | CATTGCCGACAGGATGCAG | CTCGTCATACTCCTGCTTGCTG |
| SOX11 | CGGTCAAGTGCGTGTTTCTG | CACTTTGGCGACGTTGTAGC |
| miR-221 | ACACTCCAGCTGGGAGCTACATTGTCTGCTG | CTCAACTGGTGTCGTGGA |
| U6 | CTCGCTTCGGCAGCACA | AACGCTTCACGAATTTGCGT |

## RT-qPCR assay

A High-Capacity cDNA Reverse Transcription Kit (Applied Biosystems, Waltham, MA, USA) was used to convert RNA into cDNA after RNA from treated HEPG2 and Huh7 cells was extracted using TRIzol reagent (Invitrogen, Waltham, MA, USA) according to the manufacturer's instructions. The RT-qPCR assay was performed in a 20 μL reaction mixture containing 10 μL of SYBR Green Master Mix (Applied Biosystems, Thermo Fisher Scientific), 1 μL of forward primer, 1 μL of reverse primer, 1 μL of cDNA, and 7 μL of nuclease-free water. The thermocycling conditions were as follows: 95 °C for 32 s, followed by 40 cycles of 95 °C for 6 s and 62 °C for 32 s. The sequences of the primer pairs used are listed in Table 2. SOX11 or miR-221 levels were normalized to those of the housekeeping genes β-actin or U6 and determined using the $2^{-\Delta\Delta Ct}$ method.

## Detection of cell apoptosis

The effects of CUR-BBR treatment on apoptosis were assessed using flow cytometry and an apoptosis kit (BD Biosciences, San Jose, CA, USA). Briefly, after exposure to CUR-BBR, HEPG2, and Huh7 cells were stained with Annexin V-FITC and propidium iodide according to the manufacturer's instructions. After staining, the cells were analyzed using a FACSCalibur flow cytometer (BD Biosciences, San Jose, CA, USA), and the percentage of apoptotic cells was calculated using FlowJo software (BD Biosciences, San Jose, CA, USA).

## Western blotting

Protein concentrations were determined using a bicinchoninic acid protein assay after protein removal with RIPA buffer. Equal amounts of protein were resolved by SDS-PAGE and transferred onto polyvinylidene fluoride membranes (Solarbio, Beijing, China). After blocking, the membranes were incubated with SOX11 (1:1,000, ab170916; Abcam, Cambridge, UK), cleaved caspase-3 (1:500, ab32042; Abcam, Cambridge, UK), cleaved caspase-9 (1:1,000, #9505; Cell Signaling Technology, Danvers, MA, USA), and GAPDH (1:3,000, ab8245; Abcam, Cambridge, UK) primary antibodies overnight at 4 °C. Membranes were incubated with goat anti-rabbit IgG (1:2,000, ab6721; Abcam, Cambridge, UK), and bands were detected using ECL Western blot substrate (Pierce).

## Dual-luciferase reporter assay

Potential miR-221 targets were predicted using TargetScan and miRDB databases. HEPG2 and Huh7 cells were co-transfected with an miR-221 mimic in the 3′ UTR of SOX11 in both wild-type (WT) and mutant (mut) forms. Luciferase activity was assessed after 48 h

using a dual-luciferase assay (Promega Corp., Fitchburg, WI, USA), and the Renilla luciferase activity was adjusted to firefly luciferase activity.

### Transwell assay

To perform the migration assay, both cell types were placed in the upper chamber containing the culture medium without FBS. FBS-containing culture medium was added to the lower chamber. After 48 h of incubation, both the cell lines were fixed with paraformaldehyde for 15 min and stained with crystal violet for 10 min. The experimental procedure for the invasion assay was similar to that for the migration assay, except that the upper chamber was precoated with Matrigel (Corning, Corning, NY, USA). Six randomly selected visual fields were used to count the number of migrating cells.

### Statistical analysis

All data were derived from at least three independent experiments and are expressed as mean ± SD. Statistical analysis was performed using one-way analysis of variance in conjunction with Tukey's *post-hoc* test, and Student's t-test (unpaired) was used for independent two-group analyses. Differences were considered statistically significant at $P < 0.05$.

## RESULTS

### Combined effect of CUR-BBR on the survival rate of HEPG2 and Huh7 cells and the expression of miR-221

Different concentrations of CUR and BBR (0, 3.125, 6.25, 12.5, 25, 50, 100, and 200 μM) were tested separately on both HEPG2 and Huh7 cell lines. Both CUR and BBR showed dose-dependent inhibition of HEPG2 and Huh7 cell growth according to the data measured using CCK8. At 48 h, the IC50 values of CUR on HEPG2 and Huh7 cells were 36.05 and 25.40 μM, respectively (Fig. 1A), and the IC50 values of BBR on HEPG2 and Huh7 cells were 33.76 and 28.06 μM, respectively (Fig. 1B). The dose ratios of CUR to BBR were set at 1:1, 1:2, 1:4, 2:1, and 4:1. The subsequent CCK8 determination of the IC50 values of the combined treatment is shown in Table 3. The data revealed that when CUR and BBR were combined in a ratio of 2:1, the synergistic effect was the most pronounced, as evidenced by the minimal IC50 values. Specifically, for HEPG2 cells, an optimal combined concentration of 11.38 μM (comprising 7.59 μM of CUR and 3.79 μM of BBR) was observed. Similarly, for Huh7 cells, the optimal combined concentration was 20.73 μM, which included 13.82 μM of CUR and 6.91 μM of BBR (Figs. 1C and 1D). This dosage ratio was used in the subsequent experiments. RT-qPCR results demonstrated that after treatment with CUR and BBR, the expression of miR-221 in HEPG2 cells decreased by 64% and 41%, respectively, whereas that in Huh7 cells decreased by 53% and 57%, respectively. The combined treatment with CUR and BBR exhibited a superior effect compared to either CUR or BBR treatment alone (Figs. 1E and 1F), with *P*-values of 0.0065 and <0.0001, respectively. Hence, the mechanism by which CUR+BBR inhibited HCC progression may be associated with the reduced levels of miR-221.
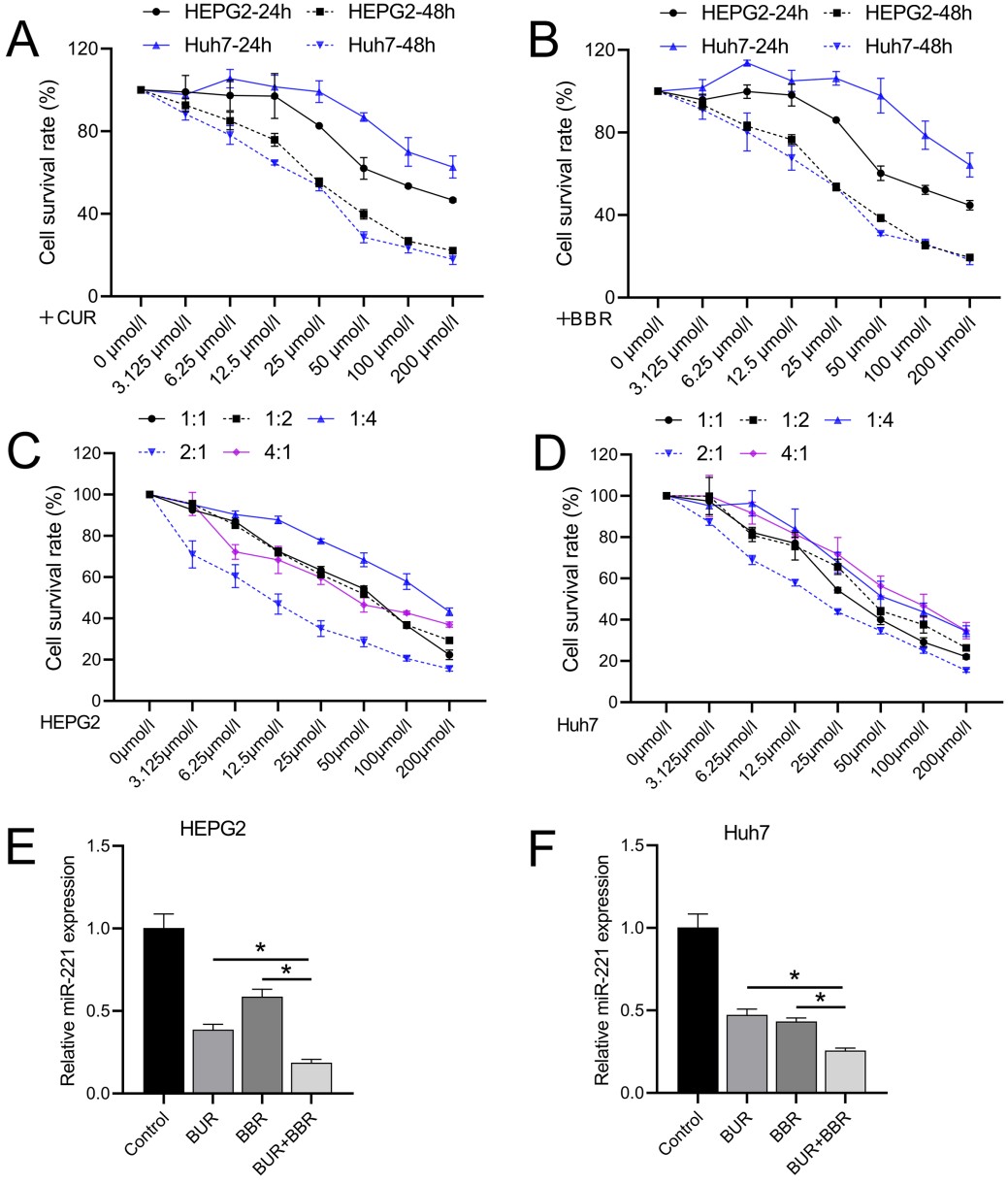

**Figure 1 The combined effect of CUR-BBR on the proliferation of hepatoma cells.** (A and B) CCK8 analysis of the toxicity of CUR and BBR at different concentrations against HEPG2 (A) or Huh7 (B) cells and their respective IC50 values. (C and D) CCK8 analysis of the toxicity of different ratios of CUR and BBR combined treatment on HEPG2 or Huh7 cells and their respective IC50 values. (E and F) RT-qPCR analysis of the effect of CUR-BBR combination therapy on the expression of miR-221 in HEPG2 (E) or Huh7 (F) cells. *$P < 0.05$.

## miR-221 reverses the combined therapeutic effect of CUR-BBR

RT-qPCR results revealed that transfection with the miR-221 mimic led to a 6.92-fold and 3.62-fold increase in miR-221 expression in HEPG2 and Huh7 cells, respectively. Conversely, transfection with the miR-221 inhibitor resulted in 83% and 86% decrease in miR-221 expression in HEPG2 and Huh7 cells, respectively (Figs. 2A and 2B). These results validated the efficacy of the miR-221 mimic/inhibitor. Furthermore, we observed

**Table 3  IC50 of CUR and BBR on HEPG2 or Huh7 cell lines.**

| CUR: BBR (HEPG2) | IC50 (μM) | CUR (μM) | BBR (μM) |
|---|---|---|---|
| 1:1 | 51.15 | 25.58 | 25.58 |
| 1:2 | 52.79 | 17.60 | 35.19 |
| 1:4 | 143.10 | 28.62 | 114.48 |
| 2:1 | 11.38 | 7.59 | 3.79 |
| 4:1 | 55.13 | 44.10 | 11.03 |
| CUR: BBR (Huh7) | | | |
| 1:1 | 36.99 | 18.50 | 18.50 |
| 1:2 | 49.66 | 16.55 | 33.11 |
| 1:4 | 72.30 | 14.46 | 57.84 |
| 2:1 | 20.73 | 13.82 | 6.91 |
| 4:1 | 55.80 | 44.64 | 11.16 |

that the miR-221 mimic enhanced the cell survival rate by 60% in HEPG2 and 62% in Huh7 cells under combined treatment with CUR+BBR. In contrast, the miR-221 inhibitor reduced the cell survival rate by 34% in HEPG2 and 49% in Huh7 cells when treated with CUR+BBR (Figs. 2C and 2D).

## CUR-BBR regulates SOX11 expression through the miR-221 pathway

TargetScan and miRDB databases revealed that SOX11 may be regulated by miR-221 among the downstream targets affected by miR-221 (Fig. 3A), affecting its own transcription and translation, thereby interfering with the development of HCC. We subsequently observed that combined CUR-BBR treatment increased the expression of SOX11 in HEPG2 and Huh7 cells by 4.78-fold and 3.51-fold, respectively, compared to treatment with CUR alone, and by 6.05-fold and 5.18-fold, respectively, compared to treatment with BBR alone (Fig. 3B). After transfection with ov-SOX11, the expression of SOX11 in HEPG2 and Huh7 cells increased by 2.10-fold and 3.34-fold, respectively, and protein levels rose by 1.41-fold and 1.55-fold, respectively, confirming the efficacy of the ov-SOX11 vector (Figs. 3C and 3D). The expression of SOX11 was negatively regulated by miR-221 (Fig. 3E); however, its overexpression had no significant effect on miR-221 expression ($P > 0.05$) (Fig. 3F).

## CUR-BBR activates apoptotic proteins *via* the miR-221/SOX11 axis

Bioinformatic analysis revealed that miR-221 has a SOX11 binding site (Fig. 4A). Subsequent dual-luciferase experiments revealed an 83% reduction in luciferase activity in the WT-SOX11 and miR-221 mimic co-transfection group compared with that in the WT-SOX11 and mimic NC co-transfection group. No difference in fluorescence activity was observed between the mut-SOX11 and miR-221 mimic or inhibitor co-transfection groups ($P > 0.05$) (Fig. 4B). miR-221 attenuated CUR-BBR-induced activation of cleaved caspase-3/9 and inhibited cell survival, and SOX11 reversed the effect of miR-221 (Figs. 4C–4E). As expected, the inhibitory effects of CUR-BBR on the migration, invasion, and apoptosis

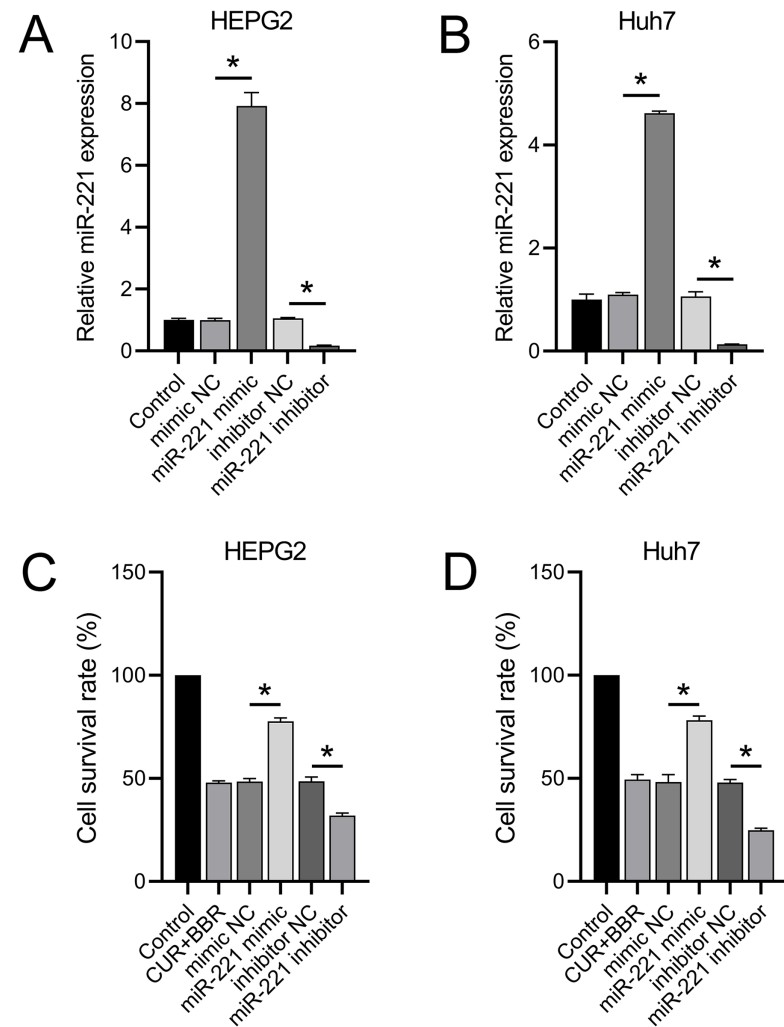

**Figure 2 miR-221 affects the combined therapeutic effect of CUR-BBR.** (A and B) RT-qPCR verified the effectiveness of miR-221 mimic and inhibitor in HEPG2 (A) and Huh7 (B) cell lines. (C and D) CCK8 analysis of the effect of miR-221 interaction with CUR-BBR on the cell survival rate of HEPG2 (C) and Huh7 (D) cells. *$P < 0.05$.

of HEPG2 and Huh7 cells were counteracted by miR-221, which was alleviated by SOX11 (Figs. 5A and 5B).

## DISCUSSION

Recent studies have suggested that traditional Chinese medicine (TCM) is a safe treatment option (*Liu et al., 2022b*). Several studies have reported the use of TCM in combination therapy (*Zhang et al., 2021b*). In this study, two TCMs, CUR and BBR, were combined to explore their mechanism of actionn in HCC treatment. Both CUR and BBR have shown significant inhibition of tumor progression in clinical settings (*Chen et al., 2020*; *Howells et al., 2019*). Existing evidence confirms that the therapeutic effect of CUR or BBR in HCC is obvious and effective (*Guo et al., 2021*) and that each can synergize with other anticancer drugs (such as CUR combined with celecoxib or BBR combined with HMQ1611 (*Abdallah*

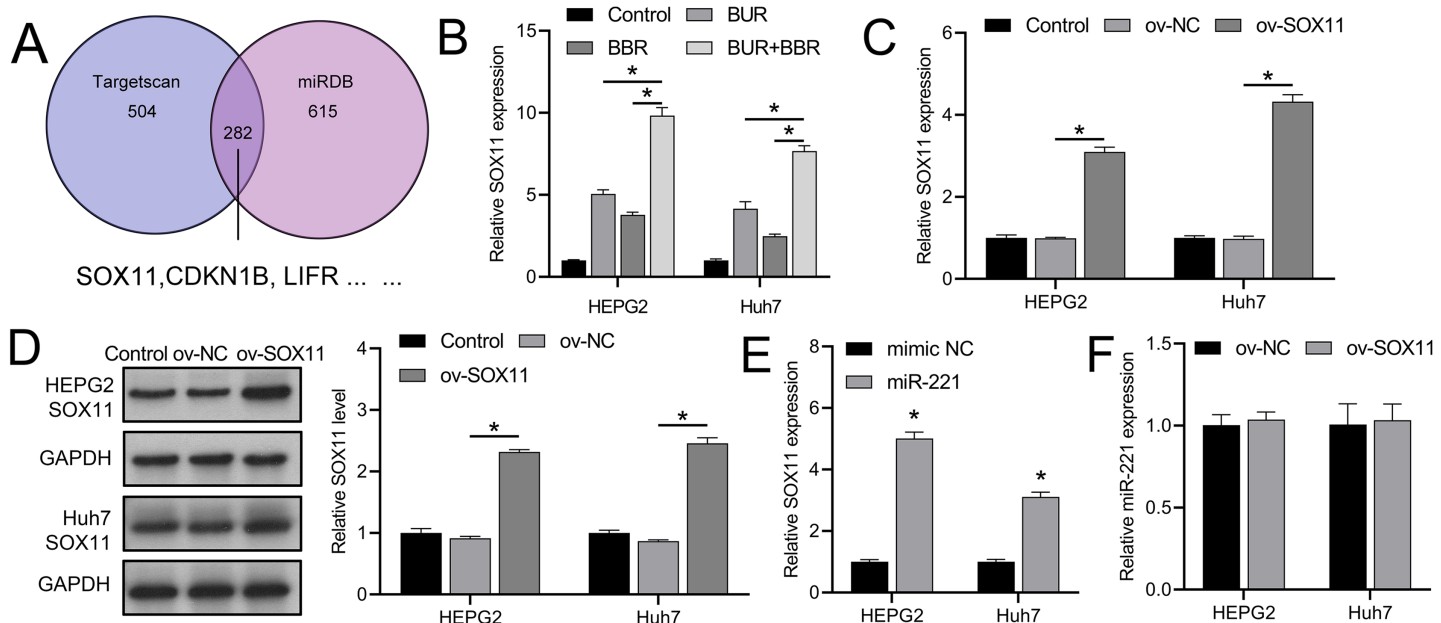

**Figure 3 CUR-BBR combination therapy promotes SOX11 expression.** (A) Targetscan and miRDB database combined to analyze potential downstream targets of miR-221. (B) RT-qPCR analysis of the effect of CUR-BBR combination therapy on the expression of SOX11 in HEPG2 and Huh7 cells. (C) RT-qPCR verified the validity of the SOX11 overexpression plasmid. (D) Western blot verified the effectiveness of overexpressing SOX11 plasmid. (E) RT-qPCR analysis of the effect of transfection of SOX11-overexpressing plasmid on the gene level of SOX11 in HEPG2 and Huh7 cells. (F) RT-qPCR analysis of the effect of transfection of SOX11-overexpressing plasmid on the gene level of miR-221 in HEPG2 and Huh7 cells. *$P < 0.05$.

*et al., 2018*; *Dai et al., 2019*)). Our study confirmed that both CUR and BBR inhibited HCC cell growth in a time- and dose-dependent manner; however, owing to their different pharmacologies, the ratios of CUR:BBR were 1:2 and 1:4. At a 4:1 ratio, the combined effect was lower than that of CUR or BBR alone. Notably, the individual effects of CUR and BBR were significantly lower than their combined effects (2:1 ratio). Therefore, in a follow-up study, we used this ratio of CUR to BBR to analyze the mechanism of action.

Recently, miR-221 has been extensively studied as a tumor-promoting factor (*Corra et al., 2021*; *Di Paolo et al., 2021*). It also acts as a biomarker of HCC and mediates its progression (*Huang et al., 2021a*). The significant downregulation of miR-221 following CUR and BBR treatment, as well as the enhanced effect of CUR-BBR combination therapy, suggests a pivotal role of miR-221 in the inhibitory effects of CUR-BBR on HCC progression, which is consistent with a previous study (*Liu et al., 2022a*). Importantly, our results showed that miR-221 overexpression partially reversed the CUR-BBR-mediated decrease in HEPG2 and Huh7 cell survival, which is the first confirmation that the anticancer effect of CUR-BBR is mediated by the miR-221 pathway.

It is well known that miRNAs act by binding to the 3′ UTR of their downstream targets (*Chen et al., 2022*). Here, we reported the direct targeting relationship of miR-221 to SOX11. Through this relationship, miR-221 regulated SOX11-mediated toxicity, apoptosis, and migration and invasion of HEPG2 and Huh7 cells. SOX11 is maintained at low levels as a tumor suppressor in HCC (*Wang et al., 2020*). SOX11 activation

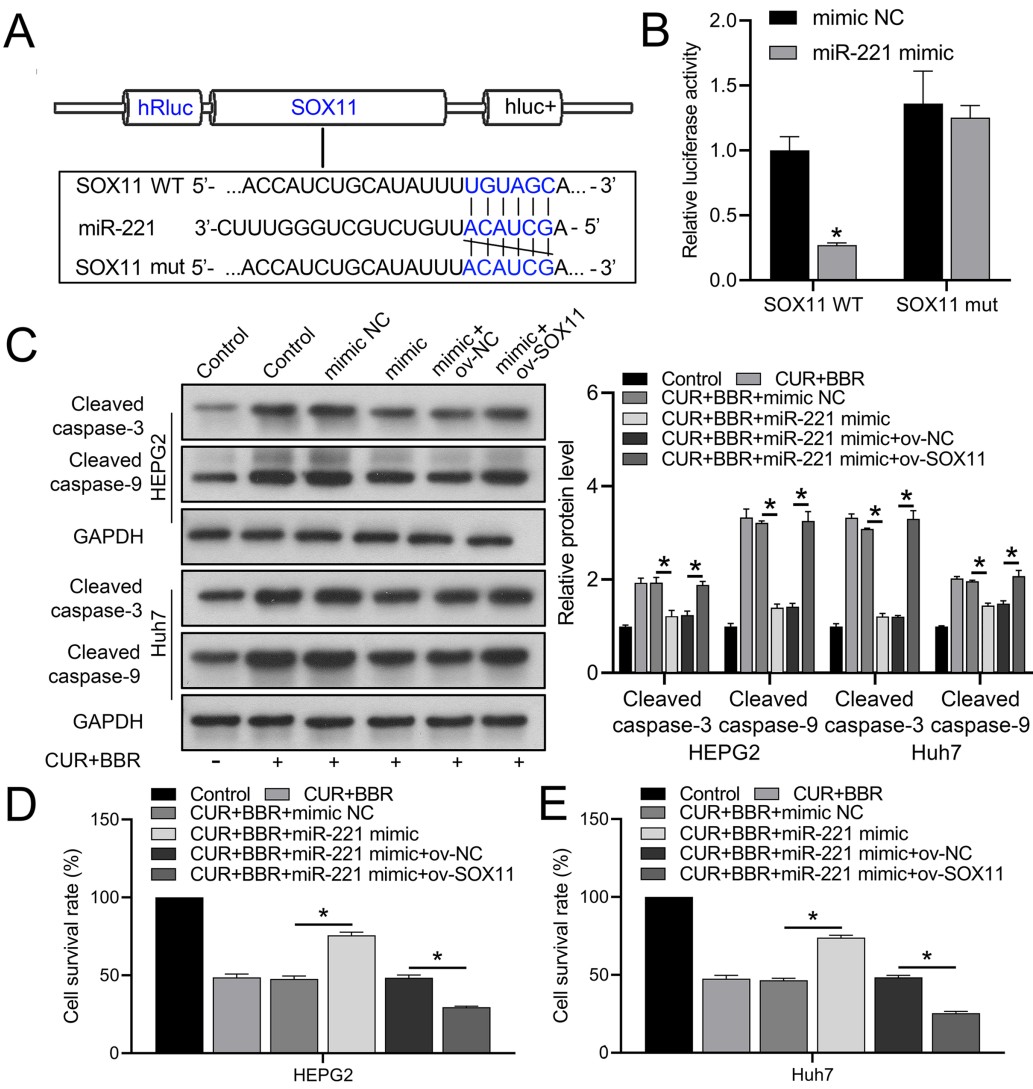

**Figure 4 miR-221 directly targets SOX11.** (A) Bioinformatics analysis of the targets of miR-221 and SOX11. (B) Dual-luciferase analysis of miR-221 binding to SOX11. (C) Western blot analysis of the effect of ov-SOX11 on the protein levels of caspase-3 and caspase-9 in the presence of CUR-BBR/miR-221. (D and E) CCK8 analysis of the effect of ov-SOX11 on the survival of HEPG2 (D) and Huh7 (E) cells in the presence of CUR-BBR/miR-221. *$P < 0.05$.

significantly promotes apoptosis and growth inhibition in HCC cells (*Liu et al., 2019a*). The activation of the pro-apoptotic proteins caspase-3/9 by CUR-BBR and the reversal of this effect by miR-221 overexpression suggest that CUR-BBR inhibits HCC progression by inducing apoptosis *via* the miR-221/SOX11 axis. Restoration of the inhibitory effect of CUR-BBR on cell growth, migration, and invasion, as well as the promotion of apoptosis in the presence of overexpressed SOX11, further supports the crucial role of the miR-221/SOX11 axis in mediating the anticancer effects of CUR-BBR. *Sun et al. (2021)* and *Tan et al. (2022)* pivotal to the progression and metastasis of HCC. Combined CUR and BBR treatment could pave the way for an alternative therapeutic strategy for HCC. Given the

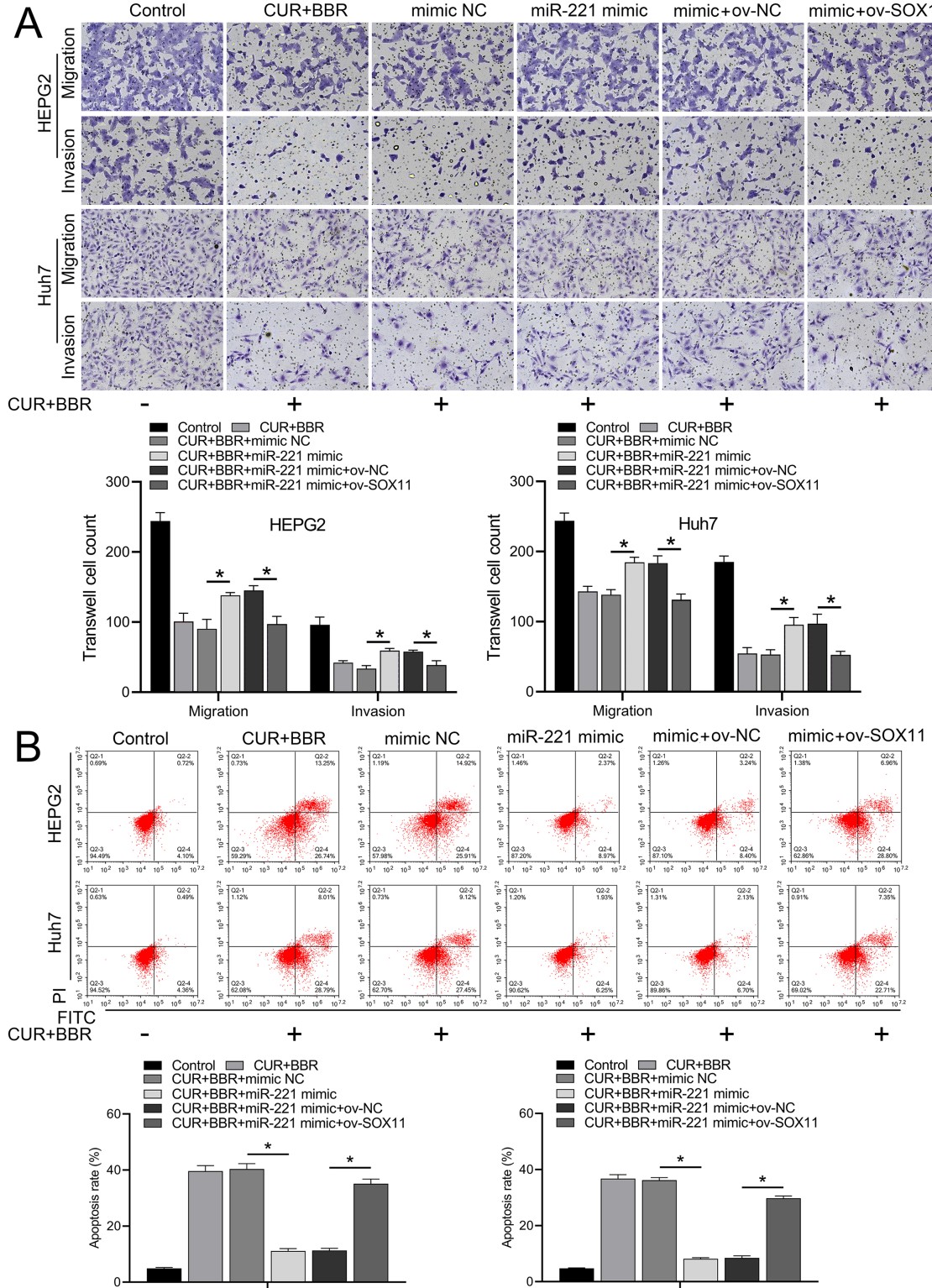

**Figure 5 CUR-BBR affects the migration/invasion and apoptosis of HEPG2 and Huh7 cells through the miR-221/SOX11 axis.** (A) Transwell analysis of the effect of ov-SOX11 on the migration and invasion of HEPG2 and Huh7 cells in the presence of CUR-BBR/miR-221. (B) Flow cytometry analysis of the effect of ov-SOX11 on apoptosis of HEPG2 and Huh7 cells in the presence of CUR-BBR/miR-221. $*P < 0.05$.

central role of miR-221, patient stratification based on miR-221 expression may further improve treatment outcomes.

This study had some limitations. Primarily, validation of these findings in broader *in vivo* animal models or clinical trials would be beneficial but was not explored in this study. Moreover, the influence of CUR and BBR on downstream signaling and interconnected molecular pathways in HCC remains largely unexplored. Future studies should address this issue by investigating the effects of CUR-BBR regimen on other associated pathways and downstream targets. Finally, we recognize the importance of translating our cellular-level discoveries into potential clinical implications, such as defining optimal dosing, foreseeing potential side effects, and evaluating patient tolerance. Although our study presented promising initial outcomes, a deeper investigation is essential to substantiate these findings and extend their potential clinical significance.

## CONCLUSIONS

Our study provided novel insights into the molecular mechanisms underlying the inhibitory effects of CUR-BBR on HCC progression. We demonstrated that CUR-BBR exerts its anticancer activity by modulating the miR-221/SOX11 axis and activating pro-apoptotic proteins. These findings not only support the potential of CUR-BBR as a promising therapeutic approach for HCC but also highlight the importance of the miR-221/SOX11 axis as a potential therapeutic target for the development of novel HCC treatments. Future studies should delve deeper into the mechanistic pathways involved and evaluate the therapeutic potential of CUR and BBR in *in vivo* HCC models.

### Funding

This study was supported by the Social Development Project for Bureau of Science and Technology of Fuzhou (Grant No. 2020-WS-108). The funders had no role in study design, data collection and analysis, decision to publish, or preparation of the manuscript.

### Grant Disclosures

The following grant information was disclosed by the authors:
Bureau of Science and Technology of Fuzhou: 2020-WS-108.

### Competing Interests

The authors declare that they have no competing interests.

### Author Contributions

- Sheng Li conceived and designed the experiments, performed the experiments, prepared figures and/or tables, authored or reviewed drafts of the article, and approved the final draft.
- Xiaoliang Cai conceived and designed the experiments, prepared figures and/or tables, and approved the final draft.

- Liang Chen analyzed the data, prepared figures and/or tables, and approved the final draft.
- Manbian Lin performed the experiments, authored or reviewed drafts of the article, and approved the final draft.
- Ziqi Zhu performed the experiments, authored or reviewed drafts of the article, and approved the final draft.
- Huihuang Xiao analyzed the data, authored or reviewed drafts of the article, and approved the final draft.
- Pingping Nie analyzed the data, authored or reviewed drafts of the article, and approved the final draft.
- Quanwen Chen performed the experiments, prepared figures and/or tables, and approved the final draft.
- Xiaoyu Yang conceived and designed the experiments, prepared figures and/or tables, and approved the final draft.

### Data Availability

The data is available at Figshare: Li, Sheng; Cai, Xiaoliang; Chen, Liang; Lin, Manbian; Zhu, Ziqi; Xiao, Huihuang; et al. (2023). The combination of curcumin and berberine inhibits the progression of hepatocellular carcinoma.zip. figshare. Dataset. https://doi.org/10.6084/m9.figshare.24047076.v1.

### Supplemental Information

Supplemental information for this article can be found online at http://dx.doi.org/10.7717/peerj.16593#supplemental-information.

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
