# Peer review of "Inhibition of hepatocellular carcinoma growth via modulation of the miR-221/SOX11 axis by curcumin and berberine"

_PeerJ, doi:10.7717/peerj.16593_

## Round 0.1 · original submission · Major Revisions

Although the three reviewers all gave positive comments on your research, they also made many comments. Please respond to each comment as requested by the reviewer.

**Language Note:** The review process has identified that the English language must be improved. PeerJ can provide language editing services - please contact us at copyediting@peerj.com for pricing (be sure to provide your manuscript number and title). Alternatively, you should make your own arrangements to improve the language quality and provide details in your response letter. – PeerJ Staff

Reviewer 1 ·

Basic reporting

1.The study provides new insights into the potential therapeutic effects of CUR and BBR on HCC. These compounds are known for their anti-cancer properties, and the study suggests that they can work together to kill HCC cells more effectively. miR-221 has been found to be a critical player in several human malignancies, including HCC, and is closely associated with poor prognosis in patients with HCC. This study has certain value, and the manuscript as a whole can benefit readers. However, it is worth noting that the language of the manuscript needs to be proofread by English speaking professionals, which is necessary.
2.Ensure consistent formatting throughout the paper, especially in the references.
3.The manuscript has a complete structure, standardized charts, and the original results can be clearly compared. However, ensure that the figures (e.g., Fig. 1A) are appropriately labeled and clear.

Experimental design

4.Elaborate on the significance of the chosen cell lines, HEPG2 and Huh7, and why they are particularly relevant for this study.
5.The mechanism of potential resistance of HCC cells to curr-bbr therapy should be considered.
6."Different concentrations of CUR and BBR (0, 3.125, 6.25, 12.5, 25, 50, 100 and 200 μM) were added to HEPG2 and Huh7 cells, respectively." For clarity, specify if each concentration was tested on both cell lines or if there was a distinction.
7.In the materials and methods section, adequate method descriptions and reagent details are necessary. Such as the RT-qPCR assay subsection, consider specifying the volume of the reaction mixture used for the PCR. Also, consider providing more details on the thermocycling conditions, especially if they differ for different primer pairs.The Western blotting subsection, mention the concentration or dilution of the secondary antibodies used.

Validity of the findings

8.When presenting results, it's crucial to provide exact values, especially when discussing statistical significance. For instance, instead of stating "significant decrease," provide the exact percentage or p-value.
9.Consider providing more detailed statistical results (e.g., exact p-values) to support the statements made.
10.Consider discussing potential side effects or toxicities associated with CUR and BBR, especially when used in combination.
11.It might be beneficial to include a "Limitations" subsection to openly address the study's constraints.

Additional comments

12.Ensure consistent use of abbreviations. Once an abbreviation is introduced (e.g., CUR for curcumin), always use the abbreviation in subsequent mentions.
13.Check for consistent formatting of chemical concentrations. For instance, ensure that "μM" is used consistently and not interchanged with "uM" or other variations.

Reviewer 2 ·

Basic reporting

Thank you for inviting me to evaluate the article titled "Inhibition of hepatocellular carcinoma growth via modulation of the miR-221/SOX11 axis by curcumin and berberine”. In the manuscript, Sheng Li et al investigated the combined effects of CUR and BBR on HEPG2 and Huh7 cell survival and miR-221 expression, synthesized miR-221 mimic/inhibitor and overexpressed SRY-box transcription factor 11(SOX11) in HEPG2 and Huh7 cells, assessed the role of miR-221/SOX11 axis in mediating the effects of CUR-BBR. The aothors demonstrated that both CUR and BBR suppressed HCC cell growth in a dose-dependent manner, with the most potent combinatorial effect observed at a 2:1 ratio. In addition, SOX11 was found to be a direct target of miR-221. CUR-BBR treatment upregulated SOX11 expression, and overexpression of SOX11 restored the inhibitory effects of CUR-BBR on cell growth, migration and invasion, and promotion of apoptosis in the presence of miR-221. They also performed that CUR-BBR activated the pro-apoptotic proteins caspase-3/9 through the miR-221/SOX11 axis and demonstrated that CUR-BBR exerts its anticancer activity by modulating the miR-221/SOX11 axis and activating pro-apoptotic proteins. This study provides novel insights into the molecular mechanisms underlying the inhibitory effects of CUR-BBR on HCC progression. contains some interesting findings that not only support the potential of CUR-BBR as a promising therapeutic approach for HCC but also highlight the importance of the miR-221/SOX11 axis as a potential therapeutic target for the development of novel HCC treatments. The authors also gived us shortcomings in this study and put forward the further research directions, which is very valuable.
However, minor revision has to be done before this manuscript could be accepted for publication in the 《PeerJ》.

Minor comments:
1 Line 76-78, can you give us more informations about the expressions of miR-221 in HCC tissues? Did you ever verify the expressions? If you are using other people’s research results, please add more references.
2 Line 83-87, SOX11 plays a cancer suppressive effect in a variety of cancers. How about the expressions and the effects of SOX11 in HCC tissues?

Experimental design

no comment

Validity of the findings

no comment

Additional comments

no comment

Reviewer 3 ·

Basic reporting

The structure of the manuscript is clear, the research purpose and experimental means are clear, the references are reasonable, and the original data can be corresponding. Then the writing and grammar of the manuscript need to be checked by a fluent English speaking professional.
The main point of the article is to investigate the synergistic effects of curcumin (CUR) and berberine (BBR) on hepatocellular carcinoma (HCC) cells and the role of miR-221 in this process. The authors found that the combined treatment of CUR and BBR decreased the survival of HCC cells, and this effect was influenced by the expression of miR-221. The article is interesting, however, the author can revise and improve it.

1. Abstract:
- The abstract provides a good overview but could benefit from a brief mention of the main methodology used.
- Consider adding a sentence about the significance or potential clinical implications of the findings.
2. Introduction:
- The introduction could benefit from a clearer statement of the research question or hypothesis at the end.
- The introduction provides a broad overview of HCC and the potential therapeutic agents. However, it might be beneficial to provide a clearer transition between the general information about HCC and the specific focus on CUR and BBR.
3. Methods:
- For the "Cell culture and transfection" subsection, specify the passage number or range of the cells used. It might be helpful to specify the seeding density of the cells for the experiments.
4. Results:
- For clarity, when discussing the effects of treatments, specify the concentrations used, especially if multiple concentrations were tested.
5. Discussion:
- The discussion could benefit from a comparison of the study's findings with previous research on CUR, BBR, and the miR-221/SOX11 axis in HCC.
- Consider discussing potential clinical implications in more depth, such as how these findings might influence treatment strategies for HCC.
6. Conclusions:
- The conclusion is concise and summarizes the main findings. Consider adding a statement about the next steps or future research directions.

Experimental design

no comment

Validity of the findings

no comment

---

## Round 0.2 · accepted · Accept

Considering the collective positive feedback from all three reviewers and the effort you have invested in addressing their comments, we are confident that the revisions you have made have significantly improved the quality and impact of the study. Thank you for choosing our journal for submitting your manuscript.

Reviewer 1 ·

Basic reporting

The author has made some revisions according to the suggestions, which basically solved my concerns, and the current version can be accepted.

Experimental design

no comment

Validity of the findings

no comment

Additional comments

no comment

Reviewer 2 ·

Basic reporting

no comment

Experimental design

no comment

Validity of the findings

no comment

Additional comments

This manuscript could be accepted for publication in the 《PeerJ》.

Reviewer 3 ·

Basic reporting

The author's revision has fulfilled some of my suggestions better, and the manuscript is much better now.

Experimental design

No further comments

Validity of the findings

No further comments

Additional comments

No further comments